# Geographical variation in rates of surgical treatment for female stress urinary incontinence in England: a national cohort study

Jil B Mamza,[1] Rebecca S Geary,[1] Dina El-Hamamsy,[2] David A Cromwell,[1] Jonathan Duckett,[3] Ash Monga,[4] Philip Toozs-Hobson,[5] Tahir Mahmood,[6] Andrew Wilson,[7] Douglas G Tincello,[7] Jan H van der Meulen,[1] Ipek Gurol Urganci[1]

JBM and RSG contributed equally.

JHvdM and IGU contributed equally.

JBM and RSG are joint first authors.

JHvdM and IGU are joint senior authors.

For numbered affiliations see end of article.

**Correspondence to**
Dr Ipek Gurol Urganci;
ipek.gurol@lshtm.ac.uk

## ABSTRACT

**Objective** To examine geographic variation in use of surgery for female stress urinary incontinence (SUI), mainly midurethral mesh tape insertions, in the English National Health Service (NHS).

**Design** National cohort study.

**Setting** NHS hospitals.

**Participants** 27 997 women aged 20 years or older who had a first SUI surgery in an English NHS Hospital between April 2013 and March 2016 and a diagnosis of SUI at the same time as the procedure.

**Methods** Multilevel Poisson regression was used to adjust for geographic differences in age, ethnicity, prevalence of long-term illness and socioeconomic deprivation.

**Primary outcome measure** Rate of surgery for SUI per 100 000 women/year at two geographic levels: Clinical Commissioning Group (CCG; n=209) and Sustainability and Transformation Partnership (STP; n=44).

**Results** The rate of surgery for SUI was 40 procedures per 100 000 women/year. Risk-adjusted rates ranged from 20 to 106 procedures per 100 000 women/year across CCGs and 24 to 69 procedures per 100 000 women/year across the STP areas. These regional differences were only partially explained by demographic characteristics as adjustment reduced variance of surgery rates by 16% among the CCGs and 35% among the STPs.

**Conclusions** Substantial geographic variation exists in the use of surgery for female SUI in the English NHS, suggesting that women in some areas are more likely to be treated compared with women with the same condition in other areas. The variation reflects differences in how national guidelines are being interpreted in the context of the ongoing debate about the safety of SUI surgery.

## INTRODUCTION

Urinary incontinence (UI) is estimated to affect 30% to 40% of women in the UK.[1 2] The condition has a significant impact on quality of life,[3] affecting physical and social activities, confidence and self-perception.[4] Stress urinary incontinence (SUI), the involuntary loss of urine with increases in abdominal pressure such as when exercising or coughing, is

### Strengths and limitations of this study

► The data used for the study include all surgical procedures performed within English National Health Service (NHS) Hospitals, reducing the risk of selection bias.

► Statistical modelling, using multilevel empirical Bayes methods was used to minimise potential estimation error problems when identifying potential outlier areas.

► Unmeasured confounding factors and differences in coding practices may have contributed to variation in surgery rates.

► This study did not account for surgical procedures performed in private hospitals. However, it is likely that at least 90% of all continence procedures in England are provided by the NHS, as the total annual spend on private healthcare in England is approximately 5% of the total annual spend on the NHS.

the most commonly diagnosed type of incontinence in women, accounting for approximately 50% of all UI diagnoses.[5] Urgency urinary incontinence (UUI) is characterised by a sudden and compelling desire to pass urine that is difficult to defer. Many women experience coexisting stress and UUI symptoms, a subtype often called mixed urinary incontinence. UI is managed at the primary care level initially.[6] Lifestyle changes may be recommended in primary care where women with UI also smoke cigarettes, report excessive fluid or caffeine consumption or are overweight or obese.[7] Surgical treatments are recommended when conservative treatments are ineffective or not tolerated.[8]

Midurethral mesh tapes were introduced in 1998 as a novel surgical treatment for female SUI.[9] A sharp rise in the use of mesh tapes to treat SUI followed, due in part to the minimally invasive nature of the procedure, with a maximum of 11 365 procedures conducted in

2009. Over the same period, the previous standard treatment for female SUI, colposuspension (a major abdominal surgery) declined from more than 3500 procedures per year to just 200.[10] However, after the peak of more than 11 000 procedures in 2008–2009, the number of mesh procedures for SUI has almost halved, falling to just 6227 by 2016-2017.[11 12] The decline in the use of mesh tapes for SUI has most likely been in response to concerns about the safety of mesh[13–15] with some women experiencing pain, dyspareunia, persistent UI and exposure or erosion.[16 17] In 2018, the use of mesh tapes to treat SUI was suspended in the National Health Service (NHS) in England, following an interim recommendation of the Independent Medicines and Medical Devices Safety Review.[18 19]

Previous studies highlighted that not all women with SUI have equitable access to appropriate incontinence care; access to continence surgery varies by age[20 21] and ethnic and socioeconomic backgrounds[22]; with evidence of variations in care for other vulnerable populations. In light of the current suspension of the use of midurethral mesh tapes, the most commonly used procedures to treat female SUI, evidence is needed regarding the use of mesh, and non-mesh, surgical continence procedures before the suspension was in place. A better understanding of geographical differences in access to surgical treatment for SUI in the English NHS between 2013 and 2016 and of the factors contributing to this variation will be informative for future policy decisions about the appropriateness of surgical treatment of female SUI.

## METHODS

### Study design, setting and definitions

This study used data from Hospital Episode Statistics (HES), a routinely collected, administrative dataset which contains records of all NHS hospital admissions in England. The cohort comprised women aged 20 years and older who had received surgical treatment for SUI between 1 April 2013 and 31 March 2016 and had an SUI diagnosis recorded at the time of the procedure. SUI surgery was defined using UK Office for Population Censuses and Surveys Classification4 codes (table 1).[23] SUI diagnosis was defined using the International Classification of Diseases Tenth Revision code: N39.3 SUI.[24] Women may have had repeat procedures in the study period, however, only the first operation was counted in calculating the rate of surgery.

### Measures

The outcome measure was rate of surgery for SUI per 100 000 women/year at two geographic levels: 209 Clinical Commissioning Group (CCG) and 44 Sustainability and Transformation Partnership (STP) areas. CCGs are statutory NHS bodies responsible for the planning and commissioning of healthcare services in a local area (average population size of about 104 000 adult women). CCG areas are grouped into 44 STP areas (average population size of about 493 000 adult women), which were set up to coordinate improvements in the delivery of NHS services.[25] Reference denominator populations were derived by aggregating the 2011 Census population counts for women aged 20 and older in lower super output areas (LSOA) that are within the respective boundaries of the CCG and STP areas. LSOAs are postcode-based hierarchical geographic units designed to improve the reporting of small area statistics in England and Wales. There are 32 844 LSOAs in England with an average population approximately 1700 people.[26]

Sociodemographic factors may explain variations in rates of surgery for SUI. We handled age as a patient-level characteristic grouped into five categories (20–39, 40–49, 50–59, 60–69 and 70+ years). Reference group was chosen as 40–49 years of age as surgery for SUI is most prevalent for this age group. Socioeconomic status, ethnicity and limiting long-term illness were CCG-level characteristics derived from 2011 Census data. For socioeconomic status, we used the averages of the national ranking of the Index of Multiple Deprivation[27] of LSOAs within each CCG and grouped the CCG averages into national quintiles ranging from 1 (most deprived CCGs) to 5 (least deprived CCGs). For ethnicity, we used the percentage of the population reporting black or ethnic minority (BME) background, and for long-term illness the percentage who reported that their day-to-day activities were limited because of a health problem or disability which has lasted, or is expected to last, at least 12 months. For each CCG, we took the averages of these percentages for LSOAs and grouped these CCG averages into national quintiles ranging from 1 (CCGs with average percentages in the lowest quintile) to 5 (highest quintile).

### Statistical analyses

We calculated the number and the unadjusted and adjusted rates per 100 000 women/year of SUI procedures overall and according to patient and regional characteristics. Incidence rate ratios (IRR) were used to represent associations between the procedure rate and regional characteristics. Multilevel Poisson regression models were used to produce empirical Bayes estimates of the unadjusted and adjusted incidence ratesfor each CCG and STP area. In addition, risk adjusted regression models were used to assess geographic variation in rates of surgery by year. The empirical Bayes estimator produces more precise results by 'pulling' the estimates for small outlier regions towards the overall mean.[28] For each area level (CCG/STP), we illustrated the amount of variation in adjusted surgery rates using maps and range plots with 99.8% credibility intervals. CCGs and STPs were marked as 'outliers' where the national average rate of surgery was not within the 99.8% credibility interval of their rates. All statistical calculations were performed using Stata V.14.

### Patient involvement

This study was supported by a steering group which included lay members and patient representatives who

**Table 1** OPCS-4 codes and counts of SUI procedures with relevant diagnosis (ICD-10) code N39.3

| OPSC-4 | Description | All operations*<br>N (%) | First operations†<br>N (%) |
|---|---|---|---|
| | *Midurethral tape insertions* | | |
| M53.3 | Introduction of tension-free vaginal tape | 16665 (57.9) | 16415 (58.6) |
| M53.6 | Introduction of transobturator tape | 8866 (30.8) | 8722 (31.2) |
| | *Injection of urethral bulking agents* | | |
| M56.3 | Endoscopic injection of inert substance into outlet of female bladder | 1628 (5.7) | 1435 (5.1) |
| | *Other abdominal/vaginal operations* | | |
| M51.1 | Abdominoperineal suspension of urethra | 32 (0.1) | 29 (0.1) |
| M51.2 | Endoscopic suspension of neck of bladder | 6 (<0.1) | 6 (<0.1) |
| M51.8 | Other specified combined abdominal and vaginal operations to support outlet of female bladder | 15 (0.1) | 13 (<0.1) |
| M51.9 | Unspecified combined abdominal and vaginal operations to support outlet of female bladder | 2 (<0.1) | 2 (<0.1) |
| M52.1 | Suprapubic sling operation | 355 (1.2) | 328 (1.2) |
| M52.2 | Retropubic suspension of neck of bladder | 78 (0.3) | 76 (0.3) |
| M52.3 | Colposuspension of neck of bladder | 587 (2.0) | 533 (1.9) |
| M52.8 | Other specified abdominal operations to support outlet of female bladder | 20 (0.1) | 15 (0.1) |
| M52.9 | Unspecified abdominal operations to support outlet of female bladder | 3 (<0.1) | 2 (<0.1) |
| M53.1 | Vaginal buttressing of urethra | 130 (0.5) | 126 (0.5) |
| M53.8 | Other specified vaginal operations to support outlet of female bladder | 302 (1.0) | 216 (0.8) |
| M53.9 | Unspecified vaginal operations to support outlet of female bladder | 5 (<0.1) | 4 (<0.1) |
| M55.2 | Implantation of artificial urinary sphincter into outlet of female bladder | 18 (0.1) | 11 (<0.1) |
| M55.6 | Insertion of retropubic device for female stress urinary incontinence NEC | 56 (0.2) | 52 (0.2) |
| M55.8 | Other specified other open operations on outlet of female bladder | 14 (<0.1) | 8 (<0.1) |
| M55.9 | Unspecified other open operations on outlet of female bladder | 0 (0) | 0 (0) |
| M58.8 | Other specified other operations on outlet of female bladder | 7 (<0.1) | 4 (<0.1) |
| M58.9 | Unspecified other operations on outlet of female bladder | 0 (0) | 0 (0) |
| | Total | 28789 | 27997 |

*For 77 episodes of care out of 28712 eligible episodes, two procedures were recorded and both are included in the overall count.
†For episodes of care where two procedures were recorded, only the more invasive or specified procedure is counted as first operation.
ICD-10, International Classification of Diseases Tenth Revision; N39.3, stress urinary incontinence ICD-10 code; OPCS, Office for Population Censuses and Surveys Classification; SUI, stress urinary incontinence.

provided input to the design of the study and interpretation of the results and contributed to the dissemination plan. The steering group met on a regular basis for the duration of the study.

## RESULTS
### Description of the cohort
Between April 2013 and March 2016, there were 33708 inpatient episodes with a surgical procedure for SUI. Of these episodes, 4996 did not satisfy the inclusion criteria, for example, because they did not have an SUI diagnosis recorded at the time of the procedure, outlined in figure 1, and 75 recorded a subsequent operation in the study period (figure 1). Of the procedures, 27997 were included in the analyses, 90% of which were midurethral mesh tape insertions (table 1). Restricting our analyses to these 27997 first SUI procedures captured >97% of all 28789 SUI procedures in the study period, and the distribution of procedure types did not vary between all and first procedures (table 1). The national annual rate of surgery was 40 procedures per 100000 women.

### Geographic variations in surgery
Figures 2A and 3A show the variation in the unadjusted empirical Bayes estimates of the procedure rates for SUI across the CCGs and STPs, with figures 2B and 3B illustrating the rates adjusted for patients' age and

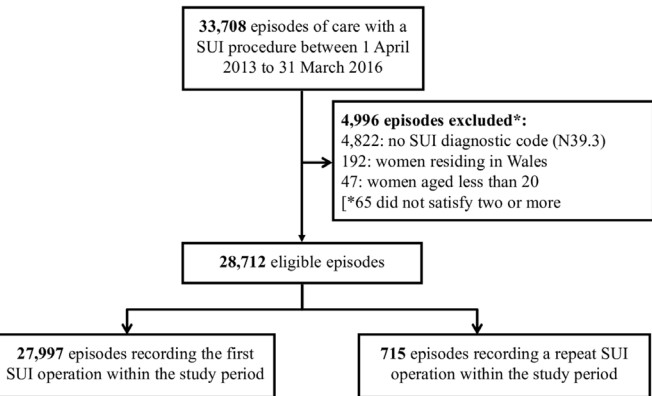

**Figure 1** Study cohort selection process.This figure is a flow diagram of the inclusion process for women who received surgical treatment for SUI in England. Data extracted from Hospital Episode Statistics 2013–2016. ICD-10, International Classification of Diseases, Tenth Revision; N39.3, stress urinary incontinence ICD-10 code; SUI, stress urinary incontinence.

the CCG-level characteristics: socioeconomic status, percentage of the population reporting BME background, and percentage with a long-term illness. Figures 2C and 3C highlight the locations of CCGs/STPs with the lowest to highest range of procedure rates in England.

The adjusted SUI procedure rates for CCGs ranged from 20 to 106 procedures compared with unadjusted rates of 11 to 120 procedures per 100 000 women/year (figure 2). Ninety-nine CCGs (47%) were marked as 'outliers' (where the national average was not within the 99.8% credibility interval of their rate). These comprised 43 CCGs (20.6%) with rates below the national average and 56 CCGs (26.8%) with rates above the national average. Risk adjustment reduced the number of CCGs marked as 'outliers' from 99 (47.4%) to 75 (36%). The SD of the CCG-level variation in adjusted rates (SD 0.27, 95% CI 0.24 to 0.30) was 16% lower than the SD of the unadjusted rates (SD 0.32, 95% CI 0.29 to 0.36).

The adjusted SUI procedure rates for STPs ranged from 24 to 69 procedures compared with unadjusted rates of 20 to 77 per 100 000 women/year (figure 3). Risk adjustment reduced the number of STPs identified as outliers from 23 (52%) to 22 (50%). The amount of variation observed declined by 35% after risk adjustment: the SD of the STP-level variation for unadjusted and adjusted model were 0.23 (95% CI 0.17 to 0.31) and 0.15 (95% CI 0.11 to 0.22), respectively.

Annual SUI procedure rates declined over the study period from 52 per 100 000 women in 2013 to 36 per 100 000 women in 2015. However, there was no evidence

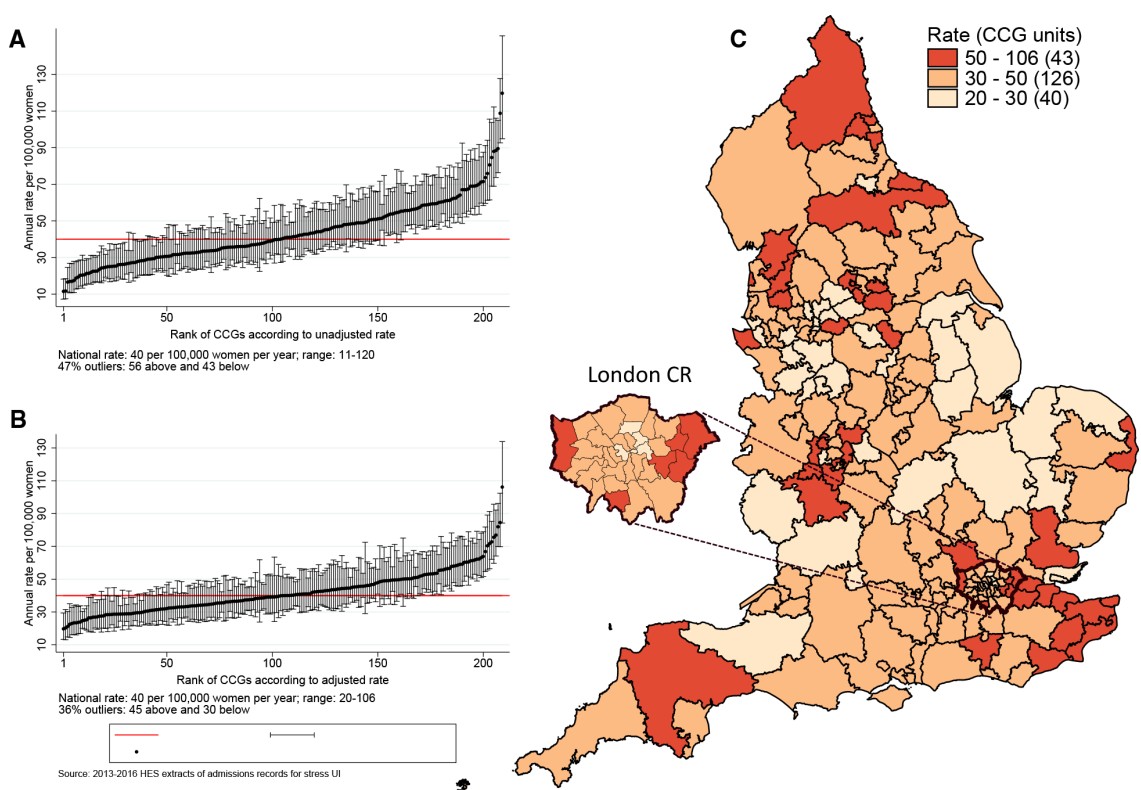

**Figure 2** CCG-level rates of stress urinary incontinence procedures between 2013 and 2016. The figure shows the EB estimated procedure rates for stress urinary incontinence. The vertical axes in (A) and (B) are EB rates. Rates in (B) are risk-adjusted for age, socioeconomic status, ethnicity and long-term illness. The numbers on the horizontal axis represent the assigned position of the CCG ranked according to rates. Geographical mapping in (C) highlight the locations of CCGs with the lowest to highest range of procedure rates in England as well as an expanded section of the London Commissioning Region. CCGs were not labelled because of space constraints. CCG, Clinical Commissioning Group; EB, empircal Bayes.

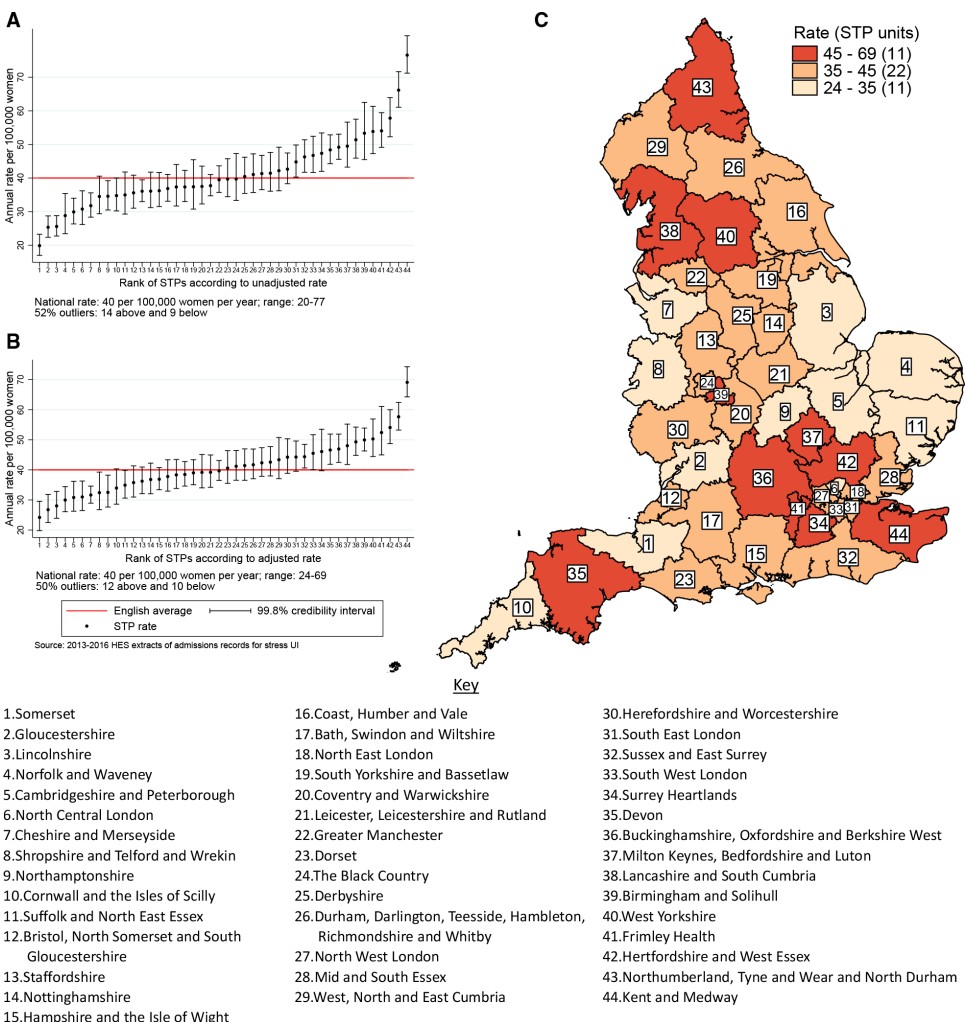

**Figure 3** STP-level rates of stress urinary incontinence procedures between 2013 and 2016. The figure shows the EB estimated procedure rates for stress urinary incontinence. The vertical axes in (A) and (B) are EB rates. Rates in (B) are risk-adjusted for age, socioeconomic status, ethnicity and long-term illness. The numbers on the horizontal axis represent the assigned position of the STP footprint ranked according to rates. Geographical mapping in (C) highlight the locations of STP footprints with the lowest to highest range of procedure rates. EB, empirical Bayes; STP, Sustainability and Transformation Partnership.

that CCG- or STP-level variation changed over time. In separate (adjusted) regression models run by year, the SD of CCG-level variation was 0.26 (95% CI 0.23 to 0.30) in 2013; 0.27 (95% CI 0.23 to 0.31) in 2014 and 0.29 (95% CI 0.25 to 0.34) in 2015. The SD of STP-level variation (adjusted model) was 0.13 (95% CI 0.08 to 0.20) in 2013, 0.17 (95% CI 0.11 to 0.25) in 2014 and 0.18 (95% CI 0.12 to 0.26) in 2015.

### Association of patient and regional characteristics with surgery rates

Table 2 shows the rates of surgery by regional characteristics. Rates were lowest for the 20–39 year age group (16 per 100 000 women/year), and highest for 40–49 year age group (84 per 100 000 women/year), declining with age beyond 50 years. Compared with the rate among women aged 40–49 years, the surgery rate for women aged 50–59 years was 20% lower (IRR 0.80, 95% CI 0.78 to 0.83), for women aged 60 to 69 years was 46% lower (IRR 0.54,

95% CI 0.52 to 0.56), and for women aged 70+ years was 69% lower (IRR 0.31, 95% CI 0.30 to 0.33).

Rates of surgery were lower for areas with higher proportions of BME populations (highest vs lowest quintile IRR 0.63, 95% CI 0.49 to 0.81). There were no differences in surgery rates according to the proportion of people with long-term limiting illness or socioeconomic deprivation at the CCG level.

### DISCUSSION
### Main findings
More than 30 000 women were admitted to NHS hospitals in England for an SUI-related surgical treatment between April 2013 and March 2016. The rate of surgery for SUI was 40 procedures per 100 000 women/year. Crude regional rates varied by a factor of 11 among the 209 CCGs from 11 to 120, and a factor of 4 among 44 STPs from 20 to 77 procedures per 100 000 women/year. These

**Table 2** Regional characteristics and their association with SUI procedure rates

| Regional factor | Scale of factor (one unit) | Procedures, n (%) | Crude rate per 100 000 women/year | Procedure rate ratio (95% CI) | P value* |
|---|---|---|---|---|---|
| **Age categories (years)** | | | | | |
| 20–39 | Age group in years | 3253 (11.6) | 15.9 | 0.18 (0.17 to 0.19) | |
| 40–49 | | 9761 (34.9) | 84.4 | Reference | <0.001 |
| 50–59 | | 7496 (26.8) | 67.5 | 0.80 (0.78 to 0.83) | |
| 60–69 | | 4352 (15.5) | 46.2 | 0.54 (0.52 to 0.56) | |
| ≥70 | | 3135 (11.2) | 26.8 | 0.31 (0.30 to 0.33) | |
| **Socioeconomic status** | | | | | |
| Most deprived | Quintile category of | 5838 (20.9) | 43.0 | Reference | 0.84 |
| More deprived | IMD ranking | 6315 (22.6) | 47.5 | 1.08 (0.93 to 1.25) | |
| Average | | 6371 (22.8) | 47.9 | 1.05 (0.89 to 1.25) | |
| Less deprived | | 5001 (17.9) | 39.9 | 1.02 (0.85 to 1.21) | |
| Least deprived | | 4472 (15.1) | 36.3 | 1.05 (0.85 to 1.29) | |
| **Black and minority ethnic population** | | | | | |
| 1: CCGs with lowest proportion | Ranked category of | 5579 (19.9) | 48.8 | Reference | 0.001 |
| 2 | Proportion of BME | 6867 (24.5) | 49.8 | 1.02 (0.89 to 1.17) | |
| 3 | Population | 6326 (22.6) | 45.7 | 1.00 (0.86 to 1.17) | |
| 4 | | 5725 (20.4) | 41.5 | 0.89 (0.75 to 1.06) | |
| 5: CCGs with highest proportion | | 3500 (12.5) | 27.2 | 0.63 (0.49 to 0.81) | |
| **Limiting long-term illness** | | | | | |
| 1: CCGs with lowest proportion | Ranked category of | 4433 (15.8) | 32.8 | Reference | 0.46 |
| 2 | Proportion of people with | 6328 (22.6) | 44.4 | 1.16 (0.99 to 1.36) | |
| 3 | Limiting illness | 4882 (17.4) | 43.7 | 1.11 (0.91 to 1.34) | |
| 4 | | 6896 (24.6) | 46.1 | 1.12 (0.91 to 1.39) | |
| 5: CCGs with highest proportion | | 5458 (19.5) | 48.9 | 1.16 (0.91 to 1.49) | |
| **Random effects estimates** | | SD† (95% CI) | | SD‡ (95% CI) | |
| STP-level variation (level 2) | | 0.23 (0.17–0.31) | | 0.15 (0.11 to 0.22) | |
| CCG-level variation (level 1) | | 0.32 (0.29–0.36) | | 0.27 (0.24 to 0.30) | |

This table describes the distribution of regional characteristics and the association between these factors and SUI procedure rates from the multilevel random-intercept Poisson regression model.
*P value obtained from likelihood ratio test.
†Unadjustedestimates.
‡Adjusted for all regional factors including ethnicity.
CCG, Clinical Commissioning Group; IMD, Index of multiple deprivation; STP, Sustainability and Transformation Plan; SUI, stress urinary incontinence.

differences were only slightly reduced when the women's age and regional characteristics were taken into account. The overall rate of SUI surgery dropped by a third over the 3-year study period, while the extent of geographic variation remained stable.

## Interpretation

This study, carried out in the English NHS, is the first national study to explore geographic variation in rates of surgical treatment for SUI. Evidence to date regarding geographic variation in benign gynaecological surgery

across England focused primarily on surgery for menorrhagia[29][30] suggesting substantial variation despite the existence of national guidelines.

We found that women's age and regional ethnicity distributions were associated with surgery rates. This may reflect differences in incontinence-related health beliefs, preferences and care seeking behaviour for older women[31] and women from various ethnic backgrounds[22] or inequitable use of surgical care.[32] Studies suggest that only around half of older people seek help for their incontinence symptoms, commonly due to the belief that it is a normal part of ageing.[33][34] In England, studies concluded that help-seeking behaviour was hindered for South-Asian women as they felt embarrassed to discuss sensitive problems, particularly with a male health professional.[35][36] Other studies in the Netherlands,[37] Sweden[38] and the USA[39] also found notable differences in preferences across women from different age groups and ethnic backgrounds.

We found that older women were less likely to have received surgical treatment for their SUI. This agrees with findings for other aspects of continence care. A national audit for continence care in the UK[20] found that deficiencies in the organisation of care and the management of UI were more pronounced for older people.[40] For example, in acute and primary care settings, older people were less likely to have a continence history or focused examination done. In secondary care, while it has been shown that surgical treatments are safe and effective in older women,[41] these procedures were used less frequently than in younger patients.[20][42]

In their work on clinical practice variation, Wennberg and colleagues emphasise three factors as possible sources of variation: clinical uncertainty about the appropriateness of care, regional differences in patients' preferences for particular treatments and differences in the capacity or supply of services.[43] In the context of SUI surgery, a part of the observed variation will reflect the ongoing debate and concerns about the safety of midurethral mesh tape procedures for women with SUI, which in 2018 led to a 'pause' in the use of mesh for the treatment of stress urinary incontinence.[13–15][18][19] It is important to note that adjustment for factors that are likely to affect patients' preferences had little impact on the geographic variation we observed for SUI surgery. However, patients' preferences will also be strongly guided by the advice received from their clinicians. The geographic areas used in this study (CCGs and STPs) are defined by NHS bodies that commission local hospital services which suggests that differences in capacity of the local healthcare system may have contributed to the observed variation.

The 'correct' rate of SUI surgery is difficult to determine, especially given the ongoing concerns about the safety of mesh tapes. However, with the observed level of geographic variation, it is likely that women in some areas were more likely to be treated compared with women with the same condition in other areas. Informed patient choice, shared decision-making and improved communication of the risks and benefits of both mesh and non-mesh procedures[44] is often proposed as a possible solution.[43] The National Institute for Health and Care Excellence (NICE), the organisation that develops clinical guidelines for the English NHS, recommends a multidisciplinary team review prior to offering invasive therapy for SUI symptoms.[8] In light of recent reviews and the current suspension of mesh tape insertions, NICE's latest draft guidance (published October 2018) also states that non-surgical options for SUI must be offered before any surgical treatment.[45] A better understanding of relevant, long-term clinical outcomes is also needed.[46][47] With the current level of uncertainty about the safety and outcomes of midurethral mesh tape insertions, it is likely that the geographic variation we observed will continue.

In England, discussions are ongoing about setting up a national prospective registry of midurethral mesh tape insertions to monitor reoperations and removals as outcomes. However, there is also a clear need to capture a wider range of clinical outcomes that are directly relevant to women, including recurrent or persistent urinary incontinence, pain and sexual dysfunction. These types of outcomes can only be collected if women themselves are actively involved in the process.

### Strengths and limitations of this study

The data used comprised information on all surgical procedures performed within the English NHS, thereby reducing the risk of selection bias. Our statistical modelling, using multilevel empirical Bayes methods allowed for minimising potential estimation error problems and taking account of area size variations for estimation of credibility intervals.[28] This approach provided a powerful and statistically robust basis for identifying potential outlier areas.

Our analyses were subject to limitations inherent to observational studies. First, unmeasured confounding factors may have contributed to variation in surgery rates. We were unable to account for potential regional variation in the average severity of the SUI problems. Second, while the overall quality of clinical information in HES has been found to be sufficiently high for research and audit purposes, inaccuracies in coding practices could have introduced some variation between geographic areas. Finally, we were unable to account for procedures done in the private sector. Although precise figures are lacking, it is likely that at least 90% of all incontinence procedures carried out in England are provided by the NHS, given that the total annual spending on private healthcare in England is about 5% of the total annual spending on the NHS.[48] While our study findings must be interpreted with caution in light of the above limitations, these are very unlikely to explain the large regional differences observed.

### CONCLUSION

We found substantial variation in the rates of surgical treatment for female SUI between geographic regions

across England. Adjusting for the women's age group and regional characteristics reduced variation only slightly. It is likely that the observed variation is in part linked to the ongoing debate about the safety of midurethral mesh tapes leading to differences in professional opinion about the appropriateness of surgical treatment for female SUI. This can only be informed by large-scale national studies monitoring long-term outcomes relevant to patients.

**Author affiliations**
[1]Department of Health Services Research and Policy, London School of Hygiene and Tropical Medicine Faculty of Public Health and Policy, London, UK
[2]Obstetrics and Gynaecology Department, University Hospitals of Leicester NHS Trust, Leicester, UK
[3]Medway NHS Foundation Trust, Gillingham, , UK
[4]University Hospital Southampton NHS Foundation Trust, Southampton, UK
[5]Birmingham Women's and Children's NHS Foundation Trust, Birmingham, UK
[6]Victoria Hospital, NHS Fife, Kirkcaldy, UK
[7]Department of Health Sciences, University of Leicester, Leicester, UK

**Acknowledgements** We would like to thank Lynn Copley and Natalie Eugene at the Clinical Effectiveness Unit of the Royal College of Surgeons of England and London School of Hygiene and Tropical Medicine for extracting the required data.

**Contributors** The study was conceived and designed by all authors. JBM and IGU organised the datasets and performed the statistical analysis, JBM wrote the first draft of the manuscript; RSG, IGU, JBM and JvdM wrote the final manuscript, with input from DEH, DC, JD, AM, PTH, TM, AW, DGT. All authors contributed to the interpretation of results and approved the final text. Joint senior authors (IGU and JvdM) and joint first authors (JBM and RSG) made an equal contribution to this study and manuscript.

**Funding** This study was supported by a grant from the National Institute for Health Research (NIHR) Health Services and Delivery Research (HS&DR) Programme (14/70/162). Independent scientific review of the proposal has been undertaken by NIHR.

**Disclaimer** The authors are solely responsible for any errors or omissions as well as the opinions expressed.

**Map disclaimer** The depiction of boundaries on the map(s) in this article do not imply the expression of any opinion whatsoever on the part of BMJ (or any member of its group) concerning the legal status of any country, territory, jurisdiction or area or of its authorities. The map(s) are provided without any warranty of any kind, either express or implied.

**Competing interests** All authors received grant funding from the National Institute for Health Research (NIHR) Health Services and Delivery Research (HS&DR) Programme (14/70/162) during the conduct of the study. AM reports personal fees from: Astellas, Pfizer, Contura and Atlantic Medical; and PTH reports personal fees from: Boston Scientifc and Ethicon, and grants and personal fees from SEP Pharma outside the submitted work.

**Patient consent for publication** Not required.

**Ethics approval** The use of Hospital Episode Statistics data for the purpose evaluations of care delivered by the NHS was approved by the Confidentiality Advisory Group of the NHS Health Research Authority (15/CAG/0148). The data are anonymised and therefore their use does not require ethical approval and individual-level patient consent.

**Provenance and peer review** Not commissioned; externally peer reviewed.

**Data availability statement** Data may be obtained from a third party and are not publicly available.

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
