## [Reviewer comments · BMJ Open]

ARTICLE DETAILS

TITLE (PROVISIONAL)	GEOGRAPHICAL VARIATION IN RATES OF SURGICAL TREATMENT FOR FEMALE STRESS URINARY INCONTINENCE IN ENGLAND: A NATIONAL COHORT STUDY
AUTHORS	Mamza, Jil; Geary, Rebecca; El-Hamamsy, Dina; Cromwell, David; Duckett, Jonathan; Monga, Ash; Tooze-Hobson, Philip; Mahmood, Tahir; Wilson, Andrew; Tincello, Doug; van der Meulen, Jan; Gurol Urganici, Ipek

VERSION 1 – REVIEW

REVIEWER	Raveen Syan Stanford University, USA
REVIEW RETURNED	21-Mar-2019

GENERAL COMMENTS	Summary: This is a retrospective cohort study examining regional variation in placement of mid-urethral mesh tape insertions from 2013-2016. The patients included were those who received treatment from NHS hospitals and had a diagnosis of SUI. This group shows that regional variability exists in rates of midurethral mesh tape placement. In addition, when considering socioeconomic, ethnic and age, patients of older age and black/ethnic populations are significantly less likely to receive midurethral mesh tape placement. High-quality statistical analysis is used to examine these relationships, and the results are very effectively displayed in geographic figures as well as clear and concise tables. This study is important as it sheds light on variations of a surgical procedure in a (primarily) centralized health care system. Though this study doesn't specifically address why it is important to know these variations, I believe it is important to know the variations that exist so as to potentially address them: for example, NHS can seek to improve accessibility to ethnic populations, educate providers in regions that have lower utilization of tape placements, etc. I think this is an excellent paper that deserves publication, following minor revisions. Introduction: - Line 19: suggest use of a different word than high (eg. Maximum)- Lines 46-51: this seems to imply that you are going to address all the evidence needed: utilization, safety and effectiveness. I would suggest either modifying this line or in the next thought (lines 51-56), state something to the effect that this study seeks to describe utilization of mid-urethral tapes.
---

	Methods:  - Why was the study restricted to 2013-2016? Please describe why you selected this time period. - Line 21: do you think this single ICD-10 code will capture all SUI patients? Any reason you didn't include additional coding for SUI? (Ex. N39.46) - Line 47: what are LSOA? (This may be obvious to British people perhaps? As an American, these descriptions are unfamiliar and more description will help make this article more relatable to international readers) Results:  - Please explain why you chose age category 40-49 as the reference (in Table 2) Discussion:  - Multiple studies in the US have examined the relationship between when FDA warnings were introduced and a likely related decline in sling. Would be worthwhile giving your opinion on how the public safety concerns may have influenced rates of sling placement during your time period (2013-2016 specifically)? More likely to have decreased over that time period? Stayed the same? - This study shows geographic variation in placement of midurethral mesh tapes, and that age and ethnicity appears to influence receiving this therapy. Why is this important information? This is not clearly spelled out in your discussion.
--	---

REVIEWER	William Gibson University of Alberta, Canada
REVIEW RETURNED	22-Mar-2019

GENERAL COMMENTS	Thank you for asking me to review this interesting and useful work, which used HES data from the NHS to examine regional variations in surgical treatment of SUI in women. The research methods, statistical analysis, and conclusions are all valid and should be published. My only suggested change is reference 16; the text refers to "Previous studies highlighted that not all women with SUI have equitable access to appropriate incontinence care; access to continence surgery varies by age(16, 17) " However, the quoted reference 16 is a review article of the future of continence care. I suspect the intended referece was Gibson, W. and A. Wagg, Are older women more likely to receive surgical treatment for stress urinary incontinence since the introduction of the mid-urethral sling? An examination of Hospital Episode Statistics data. BJOG, 2016. 123(8): p. 1386-92. Given the first author is the same, I suspect a mis-click in the author's reference manager software! (COI: I am the first author of both papers) Once this is corrected I recommend acceptance without further review.
---

REVIEWER	Debjyoti Karmakar Mercy Health ,MELBOURNE ,Australia
REVIEW RETURNED	19-Apr-2019

GENERAL COMMENTS	Dear Authors. Thanks for your important work in the current climate. I think the discussion is too long and complex , otherwise the paper itself is quite commendable
--

REVIEWER	Renly Lim University of South Australia, Australia
REVIEW RETURNED	08-May-2019

GENERAL COMMENTS	Introduction Suggest adding more information on the types of incontinence and conservative treatment options for people unfamiliar with SUI. First paragraph seems to indicate SUI only affects women, which is not true. In the third paragraph, the authors stated that "evidence is needed regarding utilisation, safety and effectiveness of mesh....." but went on to propose looking at the geographical variation. This seems like a disjointed argument why the study is needed. Can the authors explain how "use of mesh tapes to treat SUI has been suspended" is relevant to understanding geographic variation. Methods The average number of adult females in CCG and STP, and number of people who had surgery, etc are clearly presented. But, unclear what the total number of people in the dataset was? Results The authors excluded a significant number (4822 patients) who did not have an SUI diagnosis. If the patients had a mid-urethral mesh tape insertion, surely that meant the patients had SUI? Also, why were women residing in Wales and those below 20 excluded from the cohort? Discussion The discussion section is clear, concise and easy to follow. The conclusion is appropriate.
--

REVIEWER	Monica Oliveira Batista oria Associate Professor Federal University of Ceara, Nursing Department Brazil
REVIEW RETURNED	25-May-2019

GENERAL COMMENTS	Relevant study to know how the health service is reaching the needs of the population's assistance. Influences related to ethnicity and women's age were found in the spatial distribution of surgeries. It is interesting to investigate why not all hospitals are following the protocol. Would it be difficult to interpret the protocol? Training of professionals? Infrastructure? Access? 20/42 references have more than 5 years
--

VERSION 1 – AUTHOR RESPONSE

Reviewer: 1

Reviewer Name: Raveen Syan

Institution and Country: Stanford University, USA

Please state any competing interests or state 'None declared': none declared

Please leave your comments for the authors below

Summary:

This is a retrospective cohort study examining regional variation in placement of mid-urethral mesh tape insertions from 2013-2016. The patients included were those who received treatment from NHS hospitals and had a diagnosis of SUI.

This group shows that regional variability exists in rates of midurethral mesh tape placement. In addition, when considering socioeconomic, ethnic and age, patients of older age and black/ethnic populations are significantly less likely to receive midurethral mesh tape placement. High-quality statistical analysis is used to examine these relationships, and the results are very effectively displayed in geographic figures as well as clear and concise tables.

This study is important as it sheds light on variations of a surgical procedure in a (primarily) centralized health care system. Though this study doesn't specifically address why it is important to know these variations, I believe it is important to know the variations that exist so as to potentially address them: for example, NHS can seek to improve accessibility to ethnic populations, educate providers in regions that have lower utilization of tape placements, etc.

I think this is an excellent paper that deserves publication, following minor revisions.

Introduction:

- Line 19: suggest use of a different word than high (eg. Maximum)

Revised the word to maximum.

- Lines 46-51: this seems to imply that you are going to address all the evidence needed: utilization, safety and effectiveness. I would suggest either modifying this line or in the next thought (lines 51-56), state something to the effect that this study seeks to describe utilization of mid-urethral tapes.

Deleted the words safety and effectiveness. The revised sentence reads as "In light of the current suspension of the use of mid-urethral mesh tapes, the most commonly used procedures to treat female SUI, evidence is needed regarding the utilisation of mesh, and non-mesh, surgical continence procedures."

Methods:

- Why was the study restricted to 2013-2016? Please describe why you selected this time period.

In 2017, NHS England and Scottish Government published reports following concerns about the safety of mesh, and in 2018 the use of mesh tapes were suspended in the NHS England. The study period of 2013-2016 therefore covers the time when the mesh operations have started declining but still were used in the NHS, and reflects the variations of care prior to the change in policy and practice.

- Line 21: do you think this single ICD-10 code will capture all SUI patients? Any reason you didn't include additional coding for SUI? (Ex. N39.46)

Prior to this study we have conducted a coding analysis for both diagnostic and procedure coding for UI in HES. ICD10 coding in HES consists of four-digit codes, i.e. we would not identify N39.46 but rather N39.4 which is a code used primarily for urge urinary incontinence and therefore was excluded.

- Line 47: what are LSOA? (This may be obvious to British people perhaps? As an American, these descriptions are unfamiliar and more description will help make this article more relatable to international readers)

We added a short description of LSOA in the manuscript. "LSOAs are postcode-based hierarchical geographic units designed to improve the reporting of small area statistics in England and Wales. There are 32,844 LSOAs in England with an average population approximately 1,700 people".

Results:

- Please explain why you chose age category 40-49 as the reference (in Table 2)

Surgery for UI is most prevalent for this age group, therefore the reference group was chosen as 40-49. We added a sentence in methods explaining this choice. For other regional characteristics which are characterised by rankings, we chose the lowest rank to more clearly present the impact of increasing levels/ranks.

Discussion:

- Multiple studies in the US have examined the relationship between when FDA warnings were introduced and a likely related decline in sling. Would be worthwhile giving your opinion on how the public safety concerns may have influenced rates of sling placement during your time period (2013-2016 specifically)? More likely to have decreased over that time period? Stayed the same?

The annual SUI procedure rates declined over the study period from 52 per 100,000 women in 2013 to 36 per 100,000 women in 2015. However, there was no evidence that the lower- or higher-level area variations changed over time. As we focused specifically on geographic variation we haven't emphasised this in the paper.

- This study shows geographic variation in placement of midurethral mesh tapes, and that age and ethnicity appears to influence receiving this therapy. Why is this important information? This is not clearly spelled out in your discussion.

Age and ethnicity are factors that could explain the variation in care, or highlight inequities in access to or utilisation of services. We revised the sentence to include the latter with a recent reference as "This may reflect differences in incontinence-related health beliefs, preferences and care seeking behaviour for older women(29) and women from various ethnic backgrounds(20) or inequitable use of surgical care (30) ."

Reviewer: 2

Reviewer Name: William Gibson

Institution and Country: University of Alberta, Canada

Please state any competing interests or state 'None declared': None Declared

Please leave your comments for the authors below

Thank you for asking me to review this interesting and useful work, which used HES data from the NHS to examine regional variations in surgical treatment of SUI in women.

The research methods, statistical analysis, and conclusions are all valid and should be published.

My only suggested change is reference 16; the text refers to "Previous studies highlighted that not all women with SUI have equitable access to appropriate incontinence care; access to continence surgery varies by age(16, 17) "

However, the quoted reference 16 is a review article of the future of continence care. I suspect the intended reference was

Gibson, W. and A. Wagg, Are older women more likely to receive surgical treatment for stress urinary incontinence since the introduction of the mid-urethral sling? An examination of Hospital Episode Statistics data. BJOG, 2016. 123(8): p. 1386-92.

Given the first author is the same, I suspect a mis-click in the author's reference manager software! (COI: I am the first author of both papers)

Once this is corrected I recommend acceptance without further review.

Thank you for identifying the error – it is now corrected.

Reviewer: 3

Reviewer Name: Debjyoti Karmakar

Institution and Country: Mercy Health ,MELBOURNE ,Australia

Please state any competing interests or state 'None declared': None

Please leave your comments for the authors below

Dear Authors.

Thanks for your important work in the current climate. I think the discussion is too long and complex , otherwise the paper itself is quite commendable

Thank you for your comments. We made minor changes to the discussion in light of other reviewers' comments but have not modified the content significantly or shortened the text and to ensure we describe the implications, study strengths and limitations sufficiently.

Reviewer: 4

Reviewer Name: Renly Lim

Institution and Country: University of South Australia, Australia

Please state any competing interests or state 'None declared': None declared

Please leave your comments for the authors below

Introduction

Suggest adding more information on the types of incontinence and conservative treatment options for people unfamiliar with SUI. First paragraph seems to indicate SUI only affects women, which is not true.

We added in the following sentences in the first paragraph: "Stress urinary incontinence (SUI), the involuntary loss of urine with increases in abdominal pressure such as when exercising or coughing, is the most commonly diagnosed type of incontinence in women, accounting for approximately 50% of all UI diagnoses (5). Urgency urinary incontinence (UUI) is characterised by a sudden and compelling desire to pass urine that is difficult to defer. Many women experience coexisting stress and urgency UI symptoms, a sub-type often called mixed urinary incontinence. UI is managed at the primary care level initially(11). Lifestyle changes may be recommended in primary care where women with UI also smoking cigarettes, report excessive fluid or caffeine consumption or are overweight or obese(17)."

In the third paragraph, the authors stated that "evidence is needed regarding utilisation, safety and effectiveness of mesh....." but went on to propose looking at the geographical variation. This seems

like a disjointed argument why the study is needed. Can the authors explain how "use of mesh tapes to treat SUI has been suspended" is relevant to understanding geographic variation.

We revised the following sentences in this paragraph. "In light of the current suspension of the use of mid-urethral mesh tapes, the most commonly used procedures to treat female SUI, evidence is needed regarding the utilisation of mesh, and non-mesh, surgical continence procedures before the suspension was in place. A better understanding of geographical differences in access to surgical treatment for SUI in the English National Health Service (NHS) between 2013 and 2016 and of the factors contributing to this variation will be informative for future policy decisions about the appropriateness of surgical treatment of female SUI."

Methods

The average number of adult females in CCG and STP, and number of people who had surgery, etc are clearly presented. But, unclear what the total number of people in the dataset was?

All women aged 20+ residing in England was the denominator to calculate area level statistics. For the numerator, the number of SUI operations were provided in Table 1.

Results

The authors excluded a significant number (4822 patients) who did not have an SUI diagnosis. If the patients had a mid-urethral mesh tape insertion, surely that meant the patients had SUI? Also, why were women residing in Wales and those below 20 excluded from the cohort?

The restriction for SUI diagnosis were included to ensure that for non-mesh procedures we have the appropriate diagnosis code as definitions of some procedures (e.g. other abdominal / vaginal operations) are not as clearly linked to SUI. Majority of the exclusions due to missing SUI diagnoses codes were for not for MUTs but those operations which could have been undertaken for treatment of other conditions.

Wales was not included in the Hospital Episodes Statistics and we did not have access to the equivalent data (PEDW).

Census population figures are provided in 5-year age bands, therefore we started the cohort with 20, and aggregated the denominator figures to the age bands used in the study. To include all adult population (18+) we would have also needed to have the additional age band of 14-19 which did not seem appropriate and the numbers would have been very low in this group to have an impact the study results.

Discussion

The discussion section is clear, concise and easy to follow. The conclusion is appropriate.

Reviewer: 5

Reviewer Name: Monica Oliveira Batista oria

Institution and Country: Associate Professor

Federal University of Ceara, Nursing Department

Brazil

Please state any competing interests or state 'None declared': None declared

Please leave your comments for the authors below

Relevant study to know how the health service is reaching the needs of the population's assistance. Influences related to ethnicity and women's age were found in the spatial distribution of surgeries.

It is interesting to investigate why not all hospitals are following the protocol. Would it be difficult to interpret the protocol? Training of professionals? Infrastructure? Access?

This is an interesting suggestion, but we feel that it is beyond the scope of the current paper.

20/42 references have more than 5 years

VERSION 2 – REVIEW

REVIEWER	Raveen Syan University of Miami, United States of America
REVIEW RETURNED	11-Jul-2019

GENERAL COMMENTS	This is an excellent paper upon first submission, however some minor revisions were suggested to further strengthen this study. The authors have addressed these concerns very well, and it is my opinion that this paper should accepted for publication.
--

REVIEWER	Renly Lim University of South Australia
REVIEW RETURNED	05-Jul-2019

GENERAL COMMENTS	The authors have adequately addressed all concerns.
---